# Integration of “Omics”-Based Approaches in Environmental Risk Assessment to Establish Cause and Effect Relationships: A Review

**DOI:** 10.3390/toxics13090714

**Published:** 2025-08-24

**Authors:** Kirsty F. Smith, Xavier Pochon, Steven D. Melvin, Thomas T. Wheeler, Louis A. Tremblay

**Affiliations:** 1Cawthron Institute, Private Bag 2, Nelson 7042, New Zealand; kirsty.smith@cawthron.org.nz (K.F.S.); xavier.pochon@cawthron.org.nz (X.P.); tom.wheeler@cawthron.org.nz (T.T.W.); 2School of Biological Sciences, University of Auckland, Auckland 1142, New Zealand; 3Institute of Marine Science, University of Auckland, Private Bag 92019, Auckland 1142, New Zealand; 4Australian Rivers Institute, Griffith University Gold Coast, Building G51, Edmund Rice Drive, Southport, QLD 4215, Australia; s.melvin@griffith.edu.au; 5Bioeconomy Science Institute, Lincoln 7640, New Zealand

**Keywords:** genomics, epigenetics, transcriptomics, proteomics, metabolomics, biomarkers, mechanism of toxicity, evolutionary toxicology, Aotearoa New Zealand

## Abstract

Marine and freshwater environments are under increasing pressure from anthropogenic stressors. The resulting impacts on exposed ecosystems are complex and challenging to characterise. The effects may be subtle and exhibited over long time periods. Effective and robust approaches are required to characterise the physiological and genetic processes that are impacted by pollutants to assess how populations and ecosystems may be adversely affected and at risk. The objective of the review is to provide an overview of “omics” methodologies used to assess the risk of stressors on exposed biota. This review covers the development of key omics approaches and how they have been used to contribute towards improved knowledge about the effects of environmental stressors, from molecular to whole-organism and community levels of biological organisation. We provide insights into how ecotoxicogenomics approaches can be used for various aspects of environmental risk assessment by characterising toxicological mechanisms of action. This information can be used to confirm cause-and-effect relationships required to better manage risks and protect the integrity and functionality of ecosystems.

## 1. Introduction

The integrity of ecosystems is vital for economic prosperity, environmental sustainability, and human and stock health [1]. Major changes in land use result in increasing anthropogenic inputs, such as urban, agricultural, and industrial runoff, that can result in environmental decline [2,3,4]. The presence of complex mixtures of legacy pollutants and emerging organic contaminants has been described as one of the key environmental issues facing humanity [5,6,7]. It has been established that anthropogenic chemicals, also described as novel entities, are now transgressing their planetary boundary at a threshold that threatens to impact biophysical and biochemical systems needed to maintain life on Earth [8,9,10]. Pollutants can impact various physiological processes of ecosystems, including disruption to reproductive processes that can potentially lead to population decline [6,11]. Contaminants can also induce sub-lethal biological responses that can lead to adverse effects at the population and community level [12]. Better information on whether anthropogenic pressures will result in adverse effects to an ecosystem and what specific stressors or anthropogenic activities are causing the harm are some of the challenges facing environmental managers [13].

The environmental risk assessment (ERA) framework uses chemical and biological methodologies to characterise the likely or actual adverse effects of natural and anthropogenic stressors on ecosystem health [14]. Biomarkers can provide a functional measure of an organism’s response to single and complex mixtures of chemical stressors that are bioavailable in the environment, helping inform ERA processes [15]. In vitro and in vivo bioassays are useful to characterise toxicity and predict bioavailability and biological effects of contaminants [16,17,18]. Such environmental data provide multiple lines of evidence for the weight-of-evidence process to inform on the impacts of stressors on ecosystems [19]. The adverse outcome pathway (AOP) concept also integrates knowledge concerning the linkage between a molecular initiating event (MIE; e.g., a molecular interaction between a pollutant and a specific biomolecule) and an adverse health effect relevant to the ERA [20].

Ecotoxicogenomics, or the integration of omics-derived data (e.g., genomics, transcriptomics, proteomics and metabolomics) into ecotoxicology, is a developing field that is revolutionising our ability to characterise responses to environmental stressors. These methodologies, known as omics, can characterise effects across levels of biological organisation, from the molecular to the whole community [21,22,23,24,25]. Technological advances have led to unprecedented sample throughput and analytical power that are redefining the fields of ecotoxicology [26], environmental monitoring [4,27] and ERA [28]. There is potential to transform regulatory frameworks and guide decision-making processes for the management of ecosystems and remediation actions [29,30,31]. While these approaches have their advantages, clear rationales for their use in ERA platforms are required. The aim of this review was to survey the omics literature on how omics can be used to characterise pollutant hazards and explore opportunities for their inclusion into ERA frameworks (e.g., Figure 1).

## 2. Genomics

Rapid progress in the field of genomics (the study of how an individual’s genome translates into biological functions) can assist in our understanding of the impacts of chemicals on human and ecosystem health [25,32]. Various novel high-throughput sequencing (HTS) approaches have been developed (e.g., Illumina (NextSeq, HiSeq, NovaSeq), Oxford Nanopore (e.g., MinION, PromethION), as well as Pacific Biosystems (e.g., Single Molecular Real-Time (SMRT) sequencing)). The technology has advanced with a range of platforms now available that range from 100 base pair reads to over 10 kb reads, facilitating whole genome sequencing and assembly. The evolution of HTS technologies has increased sensitivity and accuracy for genomics research and enabled the development of new biomarker assays from tissue, blood, and other sample types.

These technological advances in DNA sequencing have generated more accessible sequence information for ecologically relevant, non-model species, although the functional annotation of genes in these species can still remain a challenge [33]. This is particularly important as locally relevant species can be more effective in characterising environmental risk than standard test organisms. Functional annotation identifies genes with known characteristics such as molecular function, pathways, or biological processes and relies heavily on bioinformatic techniques to identify homology to genes with a characterised function in other organisms [32]. Additionally, gene function may also be determined by experimentation and exposure analyses in lab-based assessments, allowing for a stronger mechanistic understanding of anthropogenic stressors. Hypothesis-driven chemical exposures related to specific toxicity pathways will help detect the genes involved. Accurate gene annotations are essential for informative gene expression analyses (see transcriptomics section below). There is a need to understand both the response of genomes to genetic and environmental stimuli and, conversely, the impact the environment plays on the phenotype. Genome sequencing, in combination with the tools of functional genomics, offers opportunities to develop a better understanding of phenotypic evolution and understand the regulatory pathways involved in response to toxicants [25].

Pollutants in the environment can be at extremely low concentrations but still affect genetic diversity and the overall fitness of exposed ecosystems [34]. Genetic variation can be impacted at several levels, including nucleotide polymorphisms, copy-number variants, alternative splicing, and post-transcriptional and post-translational regulation. Characterisation of the complete genotype, including the identification of causal links between toxicant pressures and changes in genetic variability at the population level, is now feasible through relatively inexpensive whole-genome sequencing of many individuals, enabled by HTS technologies [35]. Our ability to clarify the complexity of the genomic landscape that underlies the response to pollutants is within reach. Population genomics scans can reveal both directly adaptive and compensatory loci, and quantitative trait locus (QTL) mapping may further clarify the mechanistic pathways and the response of specific phenotypes [36].

## 3. Epigenomics

In association with a rapid increase in whole genome characterisation, a revolution in our understanding of alternative non-DNA nucleotide sequence-based mechanisms, collectively termed epigenetics, which also contribute to phenotypic outcomes, has occurred [28]. Sources of epigenetic variation are mechanistically diverse but all alter a given genotype’s influence on an organism’s phenotype without changes in the underlying DNA nucleotide sequences [37,38]. Molecular-level epigenetic processes can activate, reduce or completely disable the activity of particular genes by a variety of processes, including (i) methylation of cytosine residues in the DNA, (ii) remodelling of chromatin structure through chemical modification of histone proteins, and (iii) altering regulatory processes mediated by small RNA molecules [39]. Epigenetic mechanisms have been shown to be altered by environmental changes, such as temperature [40], exposure to pollutants [41,42] and toxins [43], leading to alterations in gene expression and therefore phenotypic diversity [44]. These changes can be both transgenerational (i.e., germline transmission of information) and non-transgenerational (i.e., somatic, mitotic stability) factors, both of which can potentially alter the evolutionary pathway of future generations [45]. Epigenetic variation can be viewed as an extension of phenotypic plasticity across generations (i.e., trans-generational plasticity) [46].

Environmental contaminants have been shown to induce epigenetic changes to a wide range of ecologically relevant organisms, but there is a scarcity of information on the underlying mechanisms of epigenetic processes for most types of organisms [47]. The mechanisms of epigenetic inheritance appear to be well conserved across vertebrates, but are much more variable in invertebrate taxa, albeit with data from only a small number of species (e.g., *Caenorhabditis elegans*, *Dahpnia magna*, *Crassostrea gigas*) [48,49]. The concept of a persistent epigenetic signal over multiple generations is highly relevant to biomonitoring and ecological risk assessment [50]. Targeting epigenetic signatures may also be a useful approach for identifying ecotoxicological biomarkers. Epigenetic markers on specific genes could serve as biomarkers for exposure to specific stressors or toxic outcomes. More general markers, such as global DNA methylation, might prove to be broad indicators of accumulated stress throughout an organism’s lifetime [48].

Current epigenetics literature is primarily focused on DNA methylation, as it is a universal epigenetic marker that responds to environmental cues, and there are a range of methods available for its detection and quantification [51]. The use of methods to characterise genome-wide DNA methylation patterns (e.g., methylation-sensitive amplified fragment length polymorphism; MS-AFLP [52] or enzyme-linked immunosorbent assay (ELISA)-like spectrophotometric assays [53]) has dominated the ecological epigenetics literature. These methods can be limited, however, providing only global methylation patterns indicative of a response with no information regarding changes in specific regions of the genome, including functional genes [54]. Results can also be difficult to interpret due to simultaneous changes across many genes, and, therefore, identifying patterns in toxicity becomes difficult [55,56,57]. However, they offer affordable and approachable methods to gain insight into changes in global patterns (e.g., tissue-wide, or as a function of developmental stage) of DNA methylation between discrete samples [54]. For high-resolution analysis of DNA methylation at functional loci studies, new HTS techniques, e.g., reduced representative bisulphite sequencing (RRBS; [58]), bisulphite-converted restriction site associated with DNA sequencing (bsRADseq; [59]), methylation-specific PCR (MSP; [60]), whole-genome bisulphite sequencing (WGBS; [61]), coupled with gene expression analyses (e.g., quantitative PCR, RNA-seq), are likely to provide valuable insights into epigenetic changes in affected genes associated with functional outcomes. One promising method for the systematic analysis of epigenetic processes is using CRISPR (clustered regularly interspaced short palindromic repeats)-Cas9-based tools, which can be used for epigenetic editing at specific loci [62,63,64,65]. Until recently, epigenetic studies relied only on correlations between certain epigenetic modifications and gene regulation (i.e., activation or silencing). Epigenome editing now enables direct study of the functional relevance of certain epigenetic modifications at a specific locus or a genomic region [65].

Epigenetic biomarkers may provide early warnings of adverse outcomes associated with early and/or lifelong exposures to chemicals and other environmental stressors, that may have multigenerational effects. Apical endpoints of ecotoxicological relevance, such as growth, development, and reproduction, are known to be under epigenetic regulation [66]. With further research, the study of epigenetic markers may lead to an increased understanding of mechanisms of action of contaminants, and ultimately risk [48,50].

## 4. Transcriptomics

A transcriptome comprises the complete set of RNA transcripts that are actively being expressed by the genome, in a specific cell, tissue, organ, or organism at a specific point in time. Gene expression profiling to determine changes in up- or down-regulated genes as transcribed messenger RNA (mRNA) between different conditions is a powerful and sensitive method to characterise biological responses to environmental perturbations, such as exposure to toxic chemicals [26,67]. Determining these changes in gene expression in response to a foreign substance or compound aids in the elucidation of an organism’s molecular response and description of the potential mode of action (MOA) for a possible toxicant [68]. Virtually all responses to external stressors, including toxicants, involve changes in normal patterns of gene expression. Some are direct effects, for example, compounds can bind to a transcription factor (receptor), forming a complex that modulates transcription of specific genes, whereas other responses are compensatory and reflect the response of the organism to molecular damage or cellular dysfunction [69]. Different chemicals can generate specific patterns of gene expression, which can be used to generate hypotheses on the MOA. Through transcriptomic analyses, differential gene expression can be used to identify what classes of genes are responding to chemical contamination, providing sensitive and specific mechanistic insights into the underlying observed toxicity.

Vinken [70] identified two main areas where transcriptomic data are used within the field of ecotoxicology: (i) to define molecular initiating event(s) as either stand-alone data or to complement other types of data as input into AOPs, and (ii) to provide a set of biomarkers eligible for toxicity testing and hazard identification. Transcriptomic data can serve as first-level identifiers of chemical stress, and mechanisms can be confirmed using specific single-target methods at the proteomic, metabolomic, or organism level [70].

Transcriptomic experiments in aquatic toxicology have been diverse, encompassing various techniques for detecting changes in gene expression. Long-established methods include DNA microarrays, serial analysis of gene expression (SAGE), differential display (DD) and reverse transcription quantitative polymerase chain reaction (RT-qPCR), but have been largely replaced by HTS approaches. RT-qPCR permits the accurate and sensitive quantification of mRNA levels for expression profiling of specific genes. This technique has been established as a method of choice for the quantification of mRNA transcripts of a selected gene of interest in biological samples, but it is relatively low-throughput in comparison to microarrays. The resulting data can also be highly variable and difficult to reproduce without appropriate verification and validation of both samples and primers [71]. Subsequently, other forms of PCR have been developed, including digital PCR (dPCR) [72]. dPCR technology offers the advantage of direct and independent quantification of DNA without standard curves, giving more precise and reproducible data than qPCR. As dPCR is still an emerging technique in this field, additional work is necessary to demonstrate its efficacy compared to traditional qPCR assays. (e.g., [73,74]). Targeted gene expression techniques require that the targets (and thus the represented genes) are known “a priori”, and this has been a challenge for non-model species lacking a sequenced genome or transcriptome [25].

As with genomic techniques, advances in HTS methods have revolutionised RNA analyses [75]. Parallel sequencing of millions of nucleic acid molecules can provide mechanistic insights into toxicology and provide new avenues for biomarker discovery with growing relevance for risk assessment [24]. These technologies can identify transcriptional changes and genomic targets with base pair precision in response to chemical exposure [33]. Unlike targeted methods (e.g., RT-qPCR, dPCR), transcriptome sequencing can occur without prior genomic knowledge, although accurate alignment of transcripts is greatly enhanced by genome data assemblies [76]. Statistical confidence in differential transcript expression is increased by adequate ‘depth of coverage’ of the transcriptome or read number per sample, and the detection of rare transcripts requires significantly more read coverage [77]. The expense (although this is consistently decreasing), depth of transcriptome coverage, much greater data complexity, demands for computational analysis, and computational infrastructure are important considerations in transcriptome sequencing analysis. In addition, since results are only semiquantitative, validation through RT-qPCR/dPCR is often necessary.

Transcriptomic analyses can complement traditional ecotoxicology data by providing mechanistic insight and by identifying sub-lethal organismal responses and contaminant classes underlying observed toxicity. To apply transcriptomics in the context of ERA, the endpoints measured in the field or laboratory should be repeatedly and reliably associated with population-level responses. However, transcription of mRNA is only an intermediate step in converting genetic information into proteins, not all mRNA sequences are transcribed, many under-go post transcription modifications and proteins are modified (e.g., by phosphorylation, posttranslational cleavage) before becoming physiologically active which can create a complex relationship between specific responses and biomarkers to ecological adverse events [20]. Field-based studies that use transcriptomics profiling in conjunction with health indices of relevance to the monitoring programmes represent a solid direction for future research.

## 5. Proteomics

Proteomics is the study of the complete set of proteins present in a biological system [78]. To conduct proteomic analyses, proteins are digested to peptides and then detected using sodium dodecyl sulphate (SDS) gel electrophoresis and/or liquid chromatography with high-resolution tandem mass spectrometry [79]. Identification of the cognate intact protein is achieved by querying the mass spectrometry (MS) data against a database containing the amino acid sequences of proteins [80]. Proteomics databases are typically generated from previous genome or transcriptome characterisation of the biological system [81,82]. Methods also exist for estimating the relative abundance of each of the proteins from the MS data.

Proteomes are generally more complex and dynamic than the corresponding transcriptome from the same system. This increased diversity is the result of biological modulation of the translation process, post-translational modification of proteins such as glycosylation, phosphorylation, and linkage to lipids, and variable degradation through modulation of protease activity [83]. This complexity provides a rich and highly dynamic dataset from which to evaluate the health of the biological system being analysed, such as particular organs or fluid from a sentinel species living in an ecosystem being impacted by environmental toxins.

As with other omics technical approaches, proteomics is a non-targeted approach, reducing the requirement for selection of specific biomarkers at the outset [84]. Proteomics offers the opportunity to discover novel biomarkers for which specific low-cost assays can be developed, for example, antibody-based tests [85]. However, like other omics approaches, proteomics is dependent on pre-existing databases, and this can be a limitation for non-model species. While proteomics provides the opportunity to detect and monitor biological responses to toxicants with greater selectivity of response compared with analysis of the transcriptome, this comes with the added burden of higher analytical costs and greater complexity in data analysis compared with transcriptomics [85].

A range of methodological improvements and variations have been adapted from other areas of proteomics research and applied to ecotoxicology, and the use of proteomics in the field of ecotoxicology is still being established. One challenge is the limited extent of existing protein sequence databases for many of the most important species of interest in aquatic ecosystems [86]. Meeting this challenge will require leveraging continuing advances in high-throughput sequencing and its associated data processing technologies to establish genome resources, at least in draft form, or alternatively high-quality transcriptomes, from which proteome sequence databases can be derived. Some biological responses to chemical contaminants are generic, while others may be specific to particular compounds or classes of chemicals. For example, a meta-analysis of proteomic responses in zebrafish to environmental contaminants has shown that many of the same responses occur, irrespective of the compound being used [87]. In addition, such generic biological responses to compounds have also been recognised as a challenge to delineating the mechanism of action for ecological toxins [88]. Laboratory-based experimentation using model compounds and sentinel species provides only a simplified picture of the complex interactions occurring in nature, but nevertheless provides an important starting point towards a full understanding of biological effects at the molecular level in real-world ecosystems. The application of statistical tools such as Principal Components Analysis (PCA) and improved resolution and sensitivity of detection of perturbations of the proteome in species response to environmental stress provides a means for interpreting complex responses. Ultimately, a thorough understanding of the ecotoxicology of an ecosystem may require integration of proteomics with other omics approaches, such as transcriptomics and metabolomics, as well as combining data from several sentinel species. Advances such as these may in the future produce useful biomarkers for the early detection of stress on environments, opening the potential for mitigation at an early stage, before environmental impacts become profound or irreversible.

Ecotoxicological proteomics is now moving beyond the theoretical and into the lab, and its technical viability is established. However, most studies are largely descriptive and relatively few report mechanistic insight, but continued advances in databases and knowledge of the physiology of target species will facilitate progress.

## 6. Metabolomics

Metabolomics strives for the simultaneous measurement of the comprehensive suite of small endogenous molecules occurring in a biological sample [89,90]. The term metabolomics is generally used to describe small polar metabolites, the so-called metabolome, which includes various amino acids, sugars, nucleic acids, and some polar fatty acids and their derivatives [91]. Non-polar metabolites, including various classes of lipid, also fall under the general umbrella definition of metabolomics [92]; however, lipids are in many cases categorised as a separate omics discipline (lipidomics). The distinction is worth acknowledging from the perspective of molecular structure and chemical behaviour, but less so biologically since both polar and non-polar metabolites play key roles in diverse physiological and biochemical processes [93,94]. Indeed, polar and non-polar metabolites are involved with the synthesis and degradation of larger molecules, cell signalling, and contribute toward maintaining cellular homeostasis [95,96,97]. Where genomics, transcriptomics, and proteomics (i.e., the ‘upstream’ omics cascade) reflect epigenetic, post-transcriptional or post-translational modifications that may manifest as a tangible response in an organism, metabolites offer a closer representation of what has or is happening [62,98]. This relates to metabolites being tightly connected with gene expression but also strongly influenced by immediate environmental conditions [99]. This has led to metabolomics being viewed as the omics technique most closely related to phenotype, which makes it an important focus for bridging our understanding of interactions between the genetic and environmental forces acting on an organism [100].

Metabolites are most commonly measured via liquid- or gas-chromatography paired with mass spectrometry (LC-MS and GC-MS, respectively), or using Nuclear Magnetic Resonance (NMR) spectroscopy. There are advantages and disadvantages to each analytical platform, which have been discussed and debated elsewhere (e.g., [101]) and largely condense down to greater coverage of metabolites by MS countered by better high-throughput capability, greater reproducibility, and reduced costs for NMR [102]. With the recognised strength of metabolomics spanning medical and environmental applications, considerable research has been and is being directed towards developing technological advancements for both MS and NMR analytical platforms; for MS, a big focus is on reducing analytical time and costs, and, for NMR, the focus is on improved sensitivity to extend metabolite coverage [102].

Modern untargeted tools for measuring metabolites promote greater replication, extended temporal sampling and, in the case of laboratory studies, the inclusion of concentration gradients of contaminants [103], thereby facilitating quantification of the environmental exposure and determination of non-linearity or effect thresholds. Importantly, these upfront advantages facilitating robust study design are further complemented by interpretive strength once the data are obtained, due to the highly conserved nature of metabolites and biochemical pathways across species [104], making metabolomics workflows readily transferable across a wide range of biological samples, including plants [105], animals [100,106], and cell cultures [107].

One challenge with applying metabolomics to environmental toxicology is that uncontrolled environmental variables may confound the interpretation of field-scale studies [108]. For example, metabolomics was shown to offer a robust platform to qualitatively evaluate the physiological effects of metal(loid)s in fish inhabiting natural aquatic systems, but seasonal changes to environmental conditions were also observed to modify the response [99]. This is not unique to metabolomics and has been discussed in relation to many analytical techniques [109]. Thus, a simple recommendation for field-based metabolomics, and arguably for any focus, is to include temporal sampling representing a wide range of environmental conditions to best characterise differences between sites [94,99,110]. This lack of well-defined best practices and reporting standards for metabolomics data in ecotoxicology has resulted in limited translation of studies into reporting data for risk assessments [91]. However, recent initiatives have aimed to integrate metabolomics workflows and reporting to support decision-making about chemical compounds based on their mechanistic behaviour. An international expert group has developed the MERIT initiative (MEtabolomics standaRds Initiative in Toxicology) with the aim of describing the most relevant best practice guidelines, method performance standards, and minimal reporting standards for the acquisition, processing and statistical analysis of metabolomics data within the context of regulatory toxicology [111]. The establishment of robust workflows and methodologies will enable the progressive adoption of metabolomics in regulatory decision-making [91].

## 7. Ecogenomics

Ecological genomics (ecogenomics) applies advanced molecular technologies to study organismal responses to environmental challenges in natural settings, integrating ecology and molecular biology to uncover adaptation mechanisms at various levels and interactions between environment and phenotype [112,113,114,115,116]. Meta-omics techniques, such as metagenomics, metabarcoding, metatranscriptomics, and metaproteomics, are increasingly used to analyse genetic material directly from environmental samples [117,118].

## 8. Metabarcoding or Metagenetics

Metabarcoding has rapidly become an established PCR-based method for characterising the biodiversity of ecological communities in environmental samples [4,119,120,121,122]. Metabarcoding is the application of the “DNA barcoding” approach, but it utilises HTS to amplify a target gene that allows the simultaneous identification of many taxa from a bulk or environmental sample. Metabarcoding assays can be designed to encompass broad taxonomic groups, for example, viruses, bacteria, fungi, protists, or eukaryotes [123,124,125,126,127,128], or target specific taxa, such as foraminifera, ciliates, diatoms, fish, etc. [129,130,131,132,133], depending on primer selection and the aims of the study. Three key considerations when applying metabarcoding for characterising entire communities are the taxonomic resolution provided by the target gene, the ‘universality’ of the primers (i.e., will they amplify the target gene from a wide variety of taxa), and the requirement for robust reference databases [134,135,136]. Metabarcoding is highly effective for characterising aquatic communities and holds great potential for routine biomonitoring and environmental management [30,137,138,139,140]. Compared to traditional methods, it is faster, more cost-effective, and significantly improves sample throughput, accuracy, and taxonomic range. Recent studies highlight its promising applications in marine monitoring, particularly in aquaculture, offshore drilling, and coastal development (Table 1). In particular, the strong correlations obtained between traditional indices and recently developed DNA-based metrics/indices across a variety of pollution gradients (e.g. [128,141,142,143,144] suggest that metabarcoding has the potential to complement or even replace current biomonitoring techniques in the near future (reviewed in [27,145,146,147,148]. Finally, another promising avenue is the analysis of functional metabolic pathways derived from environmental DNA bacterial 16S ribosomal DNA metabarcoding to estimate the extent of functional redundancy within and among impacted biological communities [143].

Several genetic methods that do not rely on PCR can be used to characterise biological communities through metabarcoding. Non-PCR techniques are beneficial because PCR introduces bias by preferentially binding primers to certain DNA templates, increasing the likelihood of detecting some taxa while missing others. Methods like shotgun sequencing [172] and mitochondrial enrichment via centrifugation [173] avoid PCR but require ultra-deep sequencing to recover useful data, which drives up sequencing costs, data volume, and bioinformatics complexity. As a result, these methods are not yet practical for routine non-microbial environmental biomonitoring.

A third method that avoids the need for PCR is gene enrichment, which uses short synthetic DNA probes designed from reference sequences complementary to target genes. These probes capture sequences from a shotgun library, allowing for sequencing without PCR [174,175]. However, this technique requires a robust database of reference DNA sequences to create the probes. While extensively used in biomedical research and ancient DNA studies [176,177], its potential for biomonitoring macro-invertebrates has been demonstrated [149].

## 9. Meta-Omics

Meta-omics is a relatively novel field which aims to simultaneously analyse the taxonomy and function of all species from a biological sample. Meta-omics includes all the approaches described above but applied to a community rather than a single organism (i.e., metagenomics, metatranscriptomics, metaproteomics and community metabolomics). To date, the overwhelming majority of meta-omics studies have focused on microbial communities, mostly bacteria but also fungi, archaea, viruses, and microeukaryotes (e.g., foraminifera) [110]. Medical and clinical studies of microbiomes have driven the development of meta-omics approaches, but they are increasingly being used for environmental and ecotoxicological applications (e.g., [178,179,180]).

Meta-omics enable the inference of the complex microbial community and its relationship or interactions with the environment (and/or hosts), enabling characterisation of biochemical function and systems-level microbial interactions in situ [181,182]. However, the challenges associated with omic studies are even more complex with meta-omics, including assembly and identification of DNA/RNA sequences, metabolites and proteins of non-model organisms and the huge amount of computing power required for analyses [180,183].

## 10. Omics and the Development of Biomarkers

Omics demonstrate great capability to discover new toxicological biomarkers as omics profiling enables a comprehensive characterisation of a biological sample within a single analysis [184]. These techniques have the advantage of being applicable to a large diversity of samples, and, over the past decade, high-throughput technologies, such as DNA sequencing, proteomics, and metabolomic mass spectrometry, have been used to generate large amounts of data that allow for global comparison of changes in molecular profiles that underlie cellular or organism responses. As a result, the omics-based approaches, coupled with computational and bioinformatics methods, provide unprecedented opportunities to speed up biomarker discovery and are now widely used in the field of ecotoxicology to facilitate novel biomarker development. Within the AOP framework, omics datasets also enable a more precise definition of the MIE and the selection of biomarkers relevant for assessing effects and/or exposure [20,185]. Omics data provide both gene- and pathway-level read-outs that can be used for selecting measurable and relevant biomarkers [186,187]. However, changes at the molecular level do not necessarily result in the development of adverse effects at the organism level and may reflect inherent biological variability or compensatory, reversible changes. The discrimination of these effects requires biological understanding and interpretation [187]. However, under defined conditions of cellular location, time, and biological context, these changes can provide meaningful information about biological responses to a toxicant exposure.

Although omics can still be expensive and require substantial expertise to analyse and interpret the tremendous amount of data generated, systems biology is expanding at a fast rate, and the corresponding computational tools are continuously being developed [188]. The combined use of multiple omics to perform global characterisation of genes, epigenetic marks, mRNA, proteins, and metabolites from a specific biological sample will overcome the limitation of potentially examining the uninformative biological level of organisation or biological targets (see section below) [189]. Moreover, multi-omics should be able to explain complex biological phenomena not only for a class of contaminants but for multiple stressors. Consequently, untargeted omics analyses (universal methods), at least at the outset, should be favoured to identify relevant biological pathways disrupted by contaminants, and their corresponding sets of effect biomarkers [88].

Recent international assessments of the promises and challenges of omics techniques in chemical risk assessments have concluded that the major contribution lies in the classification of substances and definition of similarity, the elucidation of the mode of action of substances, and the identification of species-specific effects and the demonstration of human health relevance [187,190]. Importantly, consistent protocols for generating, storing, processing, and interpreting omics data are required for the outcomes of omics-based studies to be reliably verified and confidently integrated into regulatory hazard and risk assessment [26,190].

## 11. Integration of Omics Approaches for Ecotoxicology

Omics technologies can generate relevant information efficiently and accurately on substance-induced perturbations in the biochemistry and physiology of cells and organisms that are associated with adverse outcomes [189]. These technologies have been adopted in the ecotoxicology research space in the hope that they would provide tools to identify an array of biomarkers of adverse effects and mechanisms of toxicity to improve substance hazard assessments and would contribute to the development of alternative methods to animal testing [187]. However, the transfer of omics methodologies into the regulatory domain remains cautious [191]. In addition to technical issues, there are complex legal issues remaining to be addressed prior to achieving the incorporation of omics data into regulatory processes [187].

A single-omics technique will detect biomolecules of one type and thus capture changes only for a small subset of the components of a particular molecular pathway. The application of single-omics analyses can lead to the identification of targeted physiological processes for specific exposure scenarios, but they have limitations for providing a systemic understanding of toxicity or adverse outcome pathways [189]. The integration of multiple omics approaches can substantially improve the characterisation of responses to a pollutant [192], and this need has been identified (e.g., [190]). Multi-omics have particular relevance in ecotoxicology for determining the association of the molecular response with certain key events of an AOP and for the refinement of AOPs [189].

To date, multi-omics approaches to toxicological research have largely consisted of two levels of omics datasets, and studies with three or more layers are rare. The selection of the molecular level for investigation strongly depends on the research question and the model system. Examples of multi-omics approaches in toxicology studies include investigation of the effects of nanomaterials [193,194], biocides [195], trace metals [196], neurotoxins [197], and forever chemicals like perfluorooctanoic acid (PFOA) [198]. Recent advancements in high-throughput omics techniques to analyse biological samples have meant that each approach can easily generate from giga- to tera-byte-sized data files. The size of resulting data files together with differences in nomenclature among data types, accurate annotation of biomolecules, sample replication and statistical power, and data sharing, represent substantial challenges, making the integration of these multi-dimensional omics data into a biologically meaningful context challenging [199]. Many powerful omics integration methods have been published and reviewed (e.g., [200,201,202]. However, no single approach fulfils all the requirements from a toxicological perspective [189] to fully integrate the four or more critical omics approaches outlined in this review. The development of these integration methods is also limited by a lack of experiments with a multi-omics design focus using up-to-date omics techniques. These types of studies will allow a more thorough evaluation of multi-omics approaches and different integration methods [189]. The application of all single omics techniques can provide important information on biological response, but these data can be strengthened even more by correlations with other omics techniques through the formation of more extensive databases, accurate annotation and assembly of datasets.

## 12. Conclusions

Omics technologies are rapidly evolving and are being applied to a range of fields. These technologies are proving useful in ecotoxicology and could have a role as part of ERA frameworks by providing insights into the hazards of pollutants across the levels of biological organisation. Aspects of the technology and data interpretation remain complex to incorporate into hazard assessment, but there is enormous potential for gaining a much more nuanced understanding of the molecular mechanisms leading to the biological and ecological effects of pollutants. Better characterisation of the MoAs can lead to the identification of biomarkers of adverse effects linked to validated and confirmed AOPs. Many omics are new approach methodologies (NAMs) with low ethical costs and could be used as part of the regulatory process for the registration of chemicals and as part of routine monitoring programmes to assess perturbations or recovery following remedial actions. Ongoing validation and standardisation of omics will continue to demonstrate their use in ecotoxicological research and environmental management.

The implementation and incorporation of omics into ERA frameworks will require ongoing dialogue across disciplines and the involvement of environmental managers and regulators. This is needed to ensure that the information generated by omics can be validated to bring added value. Interpretation of omics data requires specialised bioinformatic expertise and extensive knowledge of the molecular physiology of the organism studied. The application of omics to ecotoxicology is currently better suited to model organisms with an extensive dataset and knowledge base. The application of omics approaches to non-model organisms is more challenging and requires species-specific molecular and physiological knowledge. During the development of approaches incorporating omics, we must consider the relevance of the information they provide that is fit for purpose for the protection of target species and ecosystems.

## Figures and Tables

**Figure 1 toxics-13-00714-f001:**
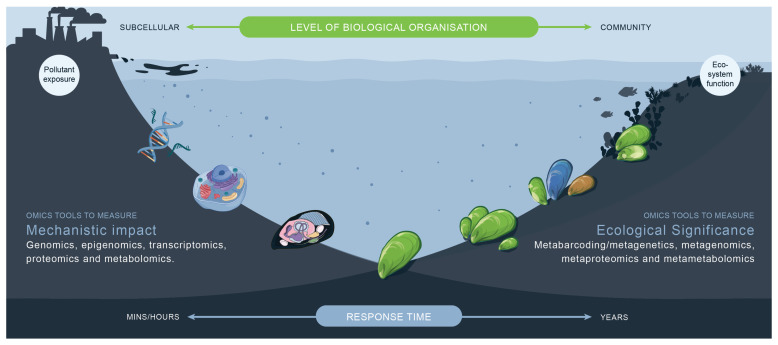
Main criteria for a framework to conduct field bioassessment must include a range of responses across levels of biological organisation. Omics approaches can generate information on toxicological mechanisms to confirm cause-and-effect relationships and better characterise the ecological impacts linked to exposures to stressors. Omics can provide information on the hazards of pollutants from molecules to whole organisms and community levels.

**Table 1 toxics-13-00714-t001:** A list of selected studies focused on marine biomonitoring using metabarcoding to assess the impacts of industry/anthropologically-derived stressors since 2015.

Stressor(s)	Biological Group(s)	Marker(s) (Marker Region)	Application(s)/Country	Reference
Salmon Aquaculture	Bacteria	16S (V4 region) ribosomal DNA (rDNA)	Monitoring benthic impacts/New Zealand	[149]
	Bacteria	16S (V4) rDNA	Monitoring benthic impacts/Norway	[132]
	Bacteria	16S (V3-V4) rDNA	Monitoring benthic impacts/international	[150]
	Foraminifera	18S (27F) rDNA	Monitoring benthic impacts/New Zealand	[151]
	Foraminifera	18S (27F) rDNA	Monitoring benthic impacts/Norway	[146]
	Foraminifera	18S (27F) rDNA	Monitoring benthic impacts/Canada	[152]
	Ciliates	18S (V9) rDNA	Monitoring benthic impacts/Scotland	[132]
	Eukaryotes	18S (V4) rDNA	Monitoring benthic impacts/Scotland	[153]
	Multi-trophic	16S (V4) rDNA, 18S (27F) rDNA, 18S (V4) rDNA	Monitoring benthic impacts/New Zealand	[142]
	Multi-trophic	16S (V3-V4) rDNA, 18S (V3-V4) rDNA	Monitoring benthic impacts/Norway	[154]
Offshore Oil and Gas	Bacteria	16S (V4) rDNA	Oil spill impacts/Gulf of Mexico	[155]
	Bacteria	16S (V3-V4) rDNA	Biocorrosive control/Brazil	[156]
	Foraminifera	18S (27F region) rDNA	Monitoring benthic impacts/New Zealand	[157]
	Prokaryotes, Eukaryotes	18S (V4-V5) rDNA	Monitoring benthic impacts/Norway	[158]
	Multi-trophic	16S (V4) rDNA, 18S (V4) rDNA	Monitoring benthic impacts/New Zealand	[159]
	Multi-trophic	18S (V1-V2; V9) rDNA, COI	Monitoring benthic impacts/North Sea (Denmark)	[160]
	Multi-trophic	18S (V1-V2; V9) rDNA	Monitoring benthic impacts/North Sea (Denmark)	[161]
	Multi-trophic	ITS2, mitochondrial 16S, COI	Monitoring old platforms/Gulf of Thailand	[162]
Coastal Development	Eukaryotes	18S (V1-V2) rDNA	Excess organic enrichment Estuaries/Australia	[163]
	Eukaryotes	COI	Metabarcoding for management/Italy	[164]
	Nematodes	18S (V1-V2) rDNA, COI	Morpho-Molecular comparison/Estuary (Portugal)	[165]
	Macroinvertebrates	COI	Monitoring network/Basque Coast (Spain)	[166]
	Teleost Fish	Mitochondrial 16S	Coastal rocky reef fish monitoring/France	[167]
	Meiofauna	18S (V1-V2) rDNA	Protected sandy beach/Germany	[168]
	Multi-trophic	18S rDNA and COI	Metabarcoding-based gAMBI index/Spain	[169]
	Multi-trophic	18S (V7) rDNA, COI	Hard-Bottom Impacts/Spain	[170]
	Multi-trophic	16S (V1-V3) rDNA, 18S (V9) rDNA	Effects of multiple stressors/Australia	[141]
	Multi-trophic	18S (V4) rDNA, COI	Tropical coastal lagoon/Mexico	[171]

rDNA: ribosomal DNA; COI: cytochrome oxidase I; ITS2: internal transcribed spacer.

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
