# Peer review of "Integration of “Omics”-Based Approaches in Environmental Risk Assessment to Establish Cause and Effect Relationships: A Review"

_toxics, 2025, doi:10.3390/toxics13090714_

Round 1
Reviewer 1 Report
Comments and Suggestions for Authors
The manuscript is logically clear and its reading is fluid, as the use of language is good. The ecotoxicogenomics methods presented are well described, with sufficient details for the reader to understand and to support the conclusions of the work, providing valuable references for environmental risk assessment. However, there are still some minor issues that need to be revised.
Line 3: Since ‘the development of these integration methods is also limited’ and ‘the technology and data interpretation remain complex to incorporate into hazard assessment’, with the current knowledge, I think the word ‘confirm’ is not used properly here. Please change the word or explain the reason for using it in more detail.
Line 548: Please change ‘Gobiomorphus cotidianus’ to ‘Gobiomorphus cotidianus’. The Latin scientific names of species should be written in italics.
Line 559: Please change ‘Innovation’ to ‘The Innovation’. The correct magazine name is ‘The Innovation’.
Line 652: This reference needs to add the doi number ‘doi:10.1007/s10646-011-0634-0’.
Please check the references to ensure that their format and content comply with the writing standards.
Author Response
Reviewer no 1
The manuscript is logically clear and its reading is fluid, as the use of language is good. The ecotoxicogenomics methods presented are well described, with sufficient details for the reader to understand and to support the conclusions of the work, providing valuable references for environmental risk assessment. However, there are still some minor issues that need to be revised.
Line 3: Since ‘the development of these integration methods is also limited’ and ‘the technology and data interpretation remain complex to incorporate into hazard assessment’, with the current knowledge, I think the word ‘confirm’ is not used properly here. Please change the word or explain the reason for using it in more detail.
Reply: Thank you for this observation. We think that “confirm” conveys the key massage but we have replaced it with “establish”.
Line 548: Please change ‘Gobiomorphus cotidianus’ to ‘Gobiomorphus cotidianus’. The Latin scientific names of species should be written in italics.
Reply: Done.
Line 559: Please change ‘Innovation’ to ‘The Innovation’. The correct magazine name is ‘The Innovation’.
Reply: Done.
Line 652: This reference needs to add the doi number ‘doi:10.1007/s10646-011-0634-0’.
Reply: Thank you, we have inserted the DOI.
Please check the references to ensure that their format and content comply with the writing standards
Reply: Thank you. We have double checked and included doi’s for those that were missing.
Reviewer 2 Report
Comments and Suggestions for Authors The review article 3818214 is the result of an analysis of the literature published in the last 25 years and confronts us with the realities and difficulties that ecotoxicology faces. Currently, the way in which research is designed and regulations are made seems to be outpaced by the increasing number of xenobiotics that reach freshwater and marine aquatic habitats, before knowing the mechanisms of toxicity and their effects manifested at the genetic and physiological levels and propagated beyond the individual level, at the population and ecosystem levels. Pollution research has gone through a long period of time to demonstrate the complexity of the aspects of pollutant toxicity, the targets affected and the magnitude of the effects, which ultimately influence biodiversity and the health of the population. The authors have managed to demonstrate that in order to objectively assess environmental risks and adopt effective measures, it is necessary to emancipate this field, by integrating new concepts, data sets derived from omics (genomics, transcriptomics, proteomics and metabolomics) and the use of modern computational analysis methods. Although this seems to be the future of ecotoxicology, the authors reveal the obstacles that come with this change of perspective: costs, limits of working methods, difficulties related to the objective analysis and interpretation of a large volume of data. I believe that the work is useful to all those working in this field of research, contributing to changing the way of thinking, the design of field activities, experimental models and the processing of results.The conclusions are synthetic, clear and explicit, supported by the arguments presented in the manuscript. Omics technologies will be useful in environmental risk assessment through information about the hazards of pollutants at different levels of organization, will clarify their modes of action and will contribute to the identification of biomarkers of adverse effects and will be used in the regulatory process for the registration of chemical substances. Despite the complexity and difficulties of interpreting the data, they will clarify the molecular mechanisms that lead to biological and ecological effects of pollutants. The manuscript was written based on the critical analysis of a number of 202 articles published between 2001-2024 in different journals, but carefully selected to support the scientific approach. I noted the scientific, rigorous, clear and synthetic style in which the manuscript was written, the attention that the authors paid to explaining new concepts and associating them with concrete examples. I have not identified any vulnerabilities in the scientific content or writing style and for this reason I have no proposals for changes to the manuscript. Taking into account the arguments presented in the report, I propose accepting the manuscript in this form.Author Response
Reviewer no 2
Reply: Thank you for the kind words and praise. We really appreciate the positivity as these articles are hard work and require cohesion across authors.
Reviewer 3 Report
Comments and Suggestions for Authors
I would like to take this opportunity to congratulate the authors of the draft "Integration of 'omics'-based approaches in environmental risk assessment to confirm cause-and-effects relationships: a review" for such an accurate review of techniques that can improve the field of ecotoxicology. Ecotoxicology uses biomarker tools to attempt to predict or infer certain degrees of xenobiotics that affect target or non-target animals.
As I consider the text to be very well written, I consider it approved.
I have just a few suggestions that could enrich it:
- provide more examples of the applications of the tools with citations
- Figure 1 is not very explanatory and could be replaced with other figures explaining the different applications of the cited omics tools.
- Topics 10 and 11 could be added to the conclusion and rewritten.
Author Response
Reviewer no 3
I would like to take this opportunity to congratulate the authors of the draft "Integration of 'omics'-based approaches in environmental risk assessment to confirm cause-and-effects relationships: a review" for such an accurate review of techniques that can improve the field of ecotoxicology. Ecotoxicology uses biomarker tools to attempt to predict or infer certain degrees of xenobiotics that affect target or non-target animals.
As I consider the text to be very well written, I consider it approved.
Reply: Thank you.
I have just a few suggestions that could enrich it:
provide more examples of the applications of the tools with citations
Reply: Thank you, we really appreciate this suggestion. We have considered but it is a question of balance. Our text is already very lengthy and covers multiple topics.
Figure 1 is not very explanatory and could be replaced with other figures explaining the different applications of the cited omics tools.
Reply: Thank you for the comment. For us we wanted to deliver a clear and easy to capture message around how omics methodologies can be used to provide insights across the different levels of biological organization. This is a very important point to make as it demonstrates that although most omics methods are considered to be at the molecular level, they can provide valuable information that can be useful even to assess health all the way to the scale of ecosystems. Again, this can be a very complex topic and we have deliberately chosen to focus on a few key messages. But we do realise that there are multiple other applications that could be better defined.
Topics 10 and 11 could be added to the conclusion and rewritten
Reply: Thank you for the suggestion. We don’t want to amend the structure of our paper as we feel that sections 10 and 11 are the key to the objective of our review. This is why, these topics require full sections to cover the key literature. The incorporation of these sections into the Conclusion would require cutting the text and that would affect the impact. As such, the Conclusion is concise and compelling
Reviewer 4 Report
Comments and Suggestions for Authors
The manuscript is a review focused on omics and its implementation in environmental risk assessment. The introduction provides the reader with the aim of the review and offers a brief general overview of the current situation in ecotoxicology.
The authors clearly explain the different omics. The review is well-documented and provides comments on most issues related to omics and the current situation. Overall, it provides a comprehensive review of the current state of Omics in ecotoxicology.
However, there are reflections to be made, but they are issues of personal consideration more than mandatory comments to include in the manuscript. The concept of “relatively inexpensive” regarding sequencing is a matter of controversy. Depending on the country, the concept of “cheap” can vary. Laboratory resources are limited, and the number of specimens that can be sequenced is also limited. In addition, in toxicology, the variability of response at different concentrations should be considered, as multiple chemicals exhibit different responses at the molecular and cellular levels, depending on them. In the end, the cost in terms of economic resources, time, and work is still high, mainly when non-model species of non-vertebrate groups are used because of the lack of good information in databases (which, in addition, have much noise because of the open approach to annotating genes by researchers).
On the other hand, the review focuses on ecotoxicology as a whole, but regarding omics, there are differences between the organisms. They can be separated into vertebrates, invertebrates, plants, and microorganisms. Vertebrates and microorganisms have more representation in databases than the other two “groups”. For plants and invertebrates, resources have been focused on species with economic or medical interest, leaving many gaps in the information that can be used for comparison (for example, in transcriptomics analysis). Therefore, many analyses performed with non-model invertebrates are biased because KEGG or GO are primarily helpful for vertebrates, but not for invertebrates. In addition, the use of omics in ecotoxicology currently focuses on stating that changes occur, rather than explaining those changes. In the articles, authors report the number of genes that change (or metabolites, proteins), but they do not perform additional analysis. In this way, genes are frequently described as CG56897 with no indication of what they are. In the case of metabolites, it is common to comment that they are related to routes that cannot even be present in the organism (e.g., related to the response in humans to a specific drug with no corresponding protein in the organism).
The authors are correct that the omics have a great deal of potential, but, as is often the case with many other methods, they are frequently used more as a means to include novelty or a fashionable method in papers rather than to be applied adequately. Regarding sequencing, there are thousands of projects in the SRA database that have been used for a single article, with no additional analysis performed. To incorporate omics into ERA, it would be necessary first to revise the database to develop a basic understanding of non-model organisms and, primarily, to assess the adequacy of the methods. For example, analyzing transcriptomes from different invertebrates reveals that about half of the sequences do not show similarity with coding genes from other species. In this half there are two groups of sequences, those that come from coding genes but correspond to 5’ UTR or 3’ UTR and those that are non-coding RNAs with a function in the cell but that are difficult to detect because the low sequence homology between species (the microRNAs and those that are small ncRNAs usually are not included in the sequencing report unless specific sequencing method was used). Additionally, it would be necessary for any agency or group of researchers to establish specific rules for the material, methods, and results sections, similar to those proposed for RT-PCR by MIQE. Currently, the trend among journals to reduce manuscript length is reflected in the poor quality of the material and methods sections, with an overuse of references that fail to adequately explain the methods employed. For Omics, it is vital to know what was done, which databases were used, with which parameters, and when. In this sense, the reviewers are also part of the problem, as they frequently do not demand a thorough description of the material and methods used.
Finally, it is worth mentioning the lack of preparation of many ecotoxicologists in relation to molecular biology. The discussion of many articles only describes the changes, and they do not attempt to explain what is happening, which pathways can be modulated, why, and what it means to the organism and the population. It is striking that changes in endocrine genes or metabolites related to energy are altered, and the discussion only indicates the change, without explaining why it is detrimental to the organism. Since many ecotoxicologists come from ecology or zoology fields, molecular and cellular biology is more a nice addition than a tool to be used and integrated with ecological endpoints. In the end, it diminishes the interest in what they have done, and the data are not used further. The lack of communication between basic researchers and ecotoxicologists is delaying the incorporation of molecular and cellular approaches and their benefits, in a similar way that occurs in medical research.
It is understood that the review focuses on the methods and the technical problems with them to be included in ERA. However, the human factor and biased databases, which lack clear criteria for researchers to use, are also delaying the potential use of omics in ERA.
Author Response
Reviewer no 4
The manuscript is a review focused on omics and its implementation in environmental risk assessment. The introduction provides the reader with the aim of the review and offers a brief general overview of the current situation in ecotoxicology.
The authors clearly explain the different omics. The review is well-documented and provides comments on most issues related to omics and the current situation. Overall, it provides a comprehensive review of the current state of Omics in ecotoxicology.
However, there are reflections to be made, but they are issues of personal consideration more than mandatory comments to include in the manuscript. The concept of “relatively inexpensive” regarding sequencing is a matter of controversy. Depending on the country, the concept of “cheap” can vary. Laboratory resources are limited, and the number of specimens that can be sequenced is also limited. In addition, in toxicology, the variability of response at different concentrations should be considered, as multiple chemicals exhibit different responses at the molecular and cellular levels, depending on them. In the end, the cost in terms of economic resources, time, and work is still high, mainly when non-model species of non-vertebrate groups are used because of the lack of good information in databases (which, in addition, have much noise because of the open approach to annotating genes by researchers).
On the other hand, the review focuses on ecotoxicology as a whole, but regarding omics, there are differences between the organisms. They can be separated into vertebrates, invertebrates, plants, and microorganisms. Vertebrates and microorganisms have more representation in databases than the other two “groups”. For plants and invertebrates, resources have been focused on species with economic or medical interest, leaving many gaps in the information that can be used for comparison (for example, in transcriptomics analysis). Therefore, many analyses performed with non-model invertebrates are biased because KEGG or GO are primarily helpful for vertebrates, but not for invertebrates. In addition, the use of omics in ecotoxicology currently focuses on stating that changes occur, rather than explaining those changes. In the articles, authors report the number of genes that change (or metabolites, proteins), but they do not perform additional analysis. In this way, genes are frequently described as CG56897 with no indication of what they are. In the case of metabolites, it is common to comment that they are related to routes that cannot even be present in the organism (e.g., related to the response in humans to a specific drug with no corresponding protein in the organism).
The authors are correct that the omics have a great deal of potential, but, as is often the case with many other methods, they are frequently used more as a means to include novelty or a fashionable method in papers rather than to be applied adequately. Regarding sequencing, there are thousands of projects in the SRA database that have been used for a single article, with no additional analysis performed. To incorporate omics into ERA, it would be necessary first to revise the database to develop a basic understanding of non-model organisms and, primarily, to assess the adequacy of the methods. For example, analyzing transcriptomes from different invertebrates reveals that about half of the sequences do not show similarity with coding genes from other species. In this half there are two groups of sequences, those that come from coding genes but correspond to 5’ UTR or 3’ UTR and those that are non-coding RNAs with a function in the cell but that are difficult to detect because the low sequence homology between species (the microRNAs and those that are small ncRNAs usually are not included in the sequencing report unless specific sequencing method was used). Additionally, it would be necessary for any agency or group of researchers to establish specific rules for the material, methods, and results sections, similar to those proposed for RT-PCR by MIQE. Currently, the trend among journals to reduce manuscript length is reflected in the poor quality of the material and methods sections, with an overuse of references that fail to adequately explain the methods employed. For Omics, it is vital to know what was done, which databases were used, with which parameters, and when. In this sense, the reviewers are also part of the problem, as they frequently do not demand a thorough description of the material and methods used.
Finally, it is worth mentioning the lack of preparation of many ecotoxicologists in relation to molecular biology. The discussion of many articles only describes the changes, and they do not attempt to explain what is happening, which pathways can be modulated, why, and what it means to the organism and the population. It is striking that changes in endocrine genes or metabolites related to energy are altered, and the discussion only indicates the change, without explaining why it is detrimental to the organism. Since many ecotoxicologists come from ecology or zoology fields, molecular and cellular biology is more a nice addition than a tool to be used and integrated with ecological endpoints. In the end, it diminishes the interest in what they have done, and the data are not used further. The lack of communication between basic researchers and ecotoxicologists is delaying the incorporation of molecular and cellular approaches and their benefits, in a similar way that occurs in medical research.
It is understood that the review focuses on the methods and the technical problems with them to be included in ERA. However, the human factor and biased databases, which lack clear criteria for researchers to use, are also delaying the potential use of omics in ERA.
Reply: Thank you so much for these very insightful and relevant reflections. It is too rare that we can engage in some collegial interactions when we write a review like this. The reviewer’s comments are so important to consider for the implementation of these methodologies. We have focused on the methods and how they can contribute to providing insights of effects on populations and ultimately communities and ecosystem functions. The relevant question when looking at a physiological response is the “so what?”. In addition to some of the references we cited, there are multiple other examples and one that comes to mind is the measurement of vitellogenin in relation to estrogenicity. How does it relate to an effect at the whole organism or community level? We would be very interested in engaging further with the reviewer as this is a topic the warrants its own paper. These are timely comments/reflections at a time when AI will be increasingly used to conduct this type of review.
We have added this text at the end of the conclusions to take into account these comments and suggestions:
The implementation and incorporation of omics into ERA frameworks will require on-going dialogue across disciplines and the involvement of environmental managers and regulators. This is needed to ensure that the information generated by omics can be validated to bring an added value. Interpretation of omics data requires specialised bioinformatic expertise and extensive knowledge of the molecular physiology of the organism studied. The application of omics to ecotoxicology is currently better suited to model organisms with an extensive dataset and knowledge base. The application of omics approaches to non-model organisms is more challenging and requires species-specific molecular and physiological knowledge. The development of approaches incorporating omics must consider the relevance of the information they provide that is fit for purpose for the protection of target species and ecosystems.